# Spatiotemporal Filtering and Noise Analysis for Regional GNSS Network in Antarctica Using Independent Component Analysis

**Wenhao Li [1], Fei Li [1,2], Shengkai Zhang [1,\*], Jintao Lei [1], Qingchuan Zhang [1] and Lexian Yuan [1]**

1   Chinese Antarctic Center of Surveying and Mapping, Wuhan University, Wuhan 430079, China; wh_li@whu.edu.cn (W.L.); fli@whu.edu.cn (F.L.); jintao.lei@whu.edu.cn (J.L.); zqc1945@126.com (Q.Z.); yuanlexian@whu.edu.cn (L.Y.)
2   State Key Laboratory of Information Engineering in Surveying, Mapping and Remote Sensing, Wuhan University, Wuhan 430079, China
\*   Correspondence: zskai@whu.edu.cn; Tel.: +86-1507-117-8326

**Abstract:** The common mode error (CME) and optimal noise model are the two most important factors affecting the accuracy of time series in regional Global Navigation Satellite System (GNSS) networks. Removing the CME and selecting the optimal noise model can effectively improve the accuracy of GNSS coordinate time series. The CME, a major source of error, is related to the spatiotemporal distribution; hence, its detrimental effects on time series can be effectively reduced through spatial filtering. Independent component analysis (ICA) is used to filter the time series recorded by 79 GPS stations in Antarctica from 2010 to 2018. After removing stations exhibiting strong local effects using their spatial responses, the filtering results of residual time series derived from principal component analysis (PCA) and ICA are compared and analyzed. The Akaike information criterion (AIC) is then used to determine the optimal noise model of the GPS time series before and after ICA/PCA filtering. The results show that ICA is superior to PCA regarding both the filter results and the consistency of the optimal noise model. In terms of the filtering results, ICA can extract multisource error signals. After ICA filtering, the root mean square (RMS) values of the residual time series are reduced by 14.45%, 8.97%, and 13.27% in the east (E), north (N), and vertical (U) components, respectively, and the associated speed uncertainties are reduced by 13.50%, 8.06% and 11.82%, respectively. Furthermore, different GNSS time series in Antarctica have different optimal noise models with different noise characteristics in different components. The main noise models are the white noise plus flicker noise (WN+FN) and white noise plus power law noise (WN+PN) models. Additionally, the spectrum index of most PN is close to that of FN. Finally, there are more stations with consistent optimal noise models after ICA filtering than there are after PCA filtering.

**Keywords:** GPS; ICA; common mode error; Antarctica; noise model

## 1. Introduction

The Global Navigation Satellite System (GNSS) velocity field, which boasts a high accuracy, constitutes an effective approach for studying regional crustal displacements; in addition, GNSS velocity solutions can validate and constrain glacial isostatic adjustment (GIA) models, which are always used as important corrections for the movements of tectonic plates, variations in the geoid, and changes in the sea level [1–6]. Studying the regional crustal displacement in Antarctica has important value as a reference for the formation and evolution of global plate tectonics in addition to creating and maintaining reference frames and monitoring the dynamics of ice and snow in

polar regions [7–15]. With the accumulation of GNSS data and the improvements in their precision, high-accuracy GNSS data have become more accessible for the study of tectonic deformation in Antarctica. GNSS coordinate time series contain both temporal and spatial correlations; the temporal correlations of GNSS coordinates can be calculated and determined using maximum likelihood estimation, and spatially correlated coordinates can be considerably reduced using a postprocessing approach, where the spatial correlations in GNSS time series are related to the spatial scale of the network. At the global scale, the predominant source of error is the reference frame, which can be reduced by a 7-parameter or 14-parameter similarity transformation [16,17]; in contrast, for a regional network, the common mode error (CME) constitutes the greatest source of error [18].

Stacking was first introduced by [18] to remove the CME to facilitate the investigation of coseismic and postseismic displacements in southern California with GPS time series. Subsequently, the lengths of time series and their baselines were adopted as weights when applying stacking filtering to regional GNSS networks [19]. Furthermore, the correlations among stations were used as weights when filtering continental-scale GPS networks [20–22]. To obtain more reliable CME estimates and to explore the physical mechanism of the CME, principal component analysis (PCA) and the Karhunen-Loeve expansion (KLE) were used to analyze 5-year GPS time series in southern California [23]. More recently, PCA, KLE, and stacking filters were employed to remove the CME from 11 GPS station time series and compared the filtering results [15]. PCA were used to filtered out the CME and evaluated its effects on the periodic signals and noise for China continuous GPS stations [24]. PCA and some modified PCA methods have been widely used in the spatiotemporal filtering of GPS time-series [25–27]. However, stacking filters assume that the CME has a uniform spatial distribution and thus cannot be applied to larger networks. In comparison, PCA is more rigorous in theory, but it is based on second-order statistics (the variance and covariance), and thus, it cannot take full advantage of higher-order statistics; in addition, the CME derived from PCA contains colored noise, which does not follow a normal distribution. Notably, the objective of PCA decomposition is to maximize the variance of each component; however, this could lead to the clustering of different physical modes within a single extracted "mathematical" mode. Recently, ICA has been used in the processing of geodetic data sets for a wide range of purposes, for example, the separation of global time-variable gravity signals [28–30], InSAR data analysis [31] and GPS time series analysis [32–35].

Alternatively, independent component analysis (ICA) were adopted for an investigation using 259 GPS stations in China [22,35], while [36] applied ICA to GPS vertical coordinate time series in Antarctica from 2010 to 2014 to explore the physical mechanism of regional filters while taking nontectonic deformation into account. Compared with PCA, ICA utilizes higher-order statistics to help differentiate statistically independent non-Gaussian signals. Therefore, considering the shortcomings of stacking and PCA filters, we use ICA to extract the CME of GNSS time series from Antarctica and analyze the applicability of both PCA and ICA from three perspectives, namely, the extraction of multisource errors, the consistency in the spatial distribution and the quality of the filtering results.

The precision of GNSS coordinate time series is influenced by not only the CME but also the noise model. Previous researchers have shown that GNSS time series contain not only white noise (WN) but also colored noise, e.g., flicker noise (FN) and random walk noise (RW) [37,38]. If the effects of colored noise are ignored, the velocity uncertainty can be overestimated by a factor of 4 or even one order of magnitude higher than the signal amplitude, leading to an incorrect physical interpretation. The maximum likelihood estimation was used to analyze the noise components of 414 GPS station time series derived from 9 different global GPS solutions and showed that the optimal noise model was WN+FN [39]. The noise models were computed for GPS time series in California and southern Nevada and implied that the optimal noise model for 50%~60% of the stations was either FN or RW, that for 25%~30% of the stations was either FN+RW or power law noise (PN), and that for 15% of the stations was either bandpass plus PN (BPPL) or first-order Gauss-Markov (FOGM) plus RW [40]. 12 GPS stations were examined in Hong Kong and stated that the optimal noise model of the corresponding GPS time series after PCA filtering was variable WN (VM) plus FN [41]. The variations in the noise at

12 GPS stations were analyzed after removing the surface mass load in China and indicated that the optimal model for 64% of the stations was FN+WN, that for 21% of the stations was BPPL+WN, that for 9% of the stations was PL+WN, and that for only 3% of the stations was either FOGM+RW+WN or FN+RW [42]. The magnitudes of WN and WN+FN+RW were estimated in Antarctica peninsula using 8 GPS stations before and after PCA filtering [43].

For regions with a vast spatial area and complex terrain such as Antarctica, the system errors, random errors and local terrain effects differ substantially among the stations in a network; the same is true for the optimal noise model, and thus, it is not sufficient to reasonably and effectively model all GNSS station time series with only one noise model. In this paper, we focus on the need to specify the optimal noise models for the GNSS stations in Antarctica. To this end, five noise models are adopted: WN+PN, WN+RW, WN+FN, WN+FN+RW, and WN+RW plus generalized Gauss-Markov (GGM).

Recently, the atmospheric load and nontide effects were computed in the CME of 53 stations from 2010 to 2014 and computed the amplitude and velocity at GPS stations with the WN+PL model before and after filtering GPS data from Antarctica [36]. First, considering the technological developments in the accumulation and accuracy of GNSS data, we perform factor analysis on GNSS time series to further explore the validity of removing the CME, investigate the applicability of a regional filter by means of ICA in Antarctica, and ascertain the optimal noise model and model variety before and after filtering the GNSS time series. Second, we briefly introduce ICA to filter the time series derived from 79 GPS stations in Antarctica from 2010 to 2018 and then compare and analyze the filtering results derived from PCA and ICA from 3 perspectives: the extraction of multisource errors in the signals, the consistency of the spatial distribution and the filter performance. Finally, the Akaike information criterion (AIC, [44,45]) is used to determine the optimal noise model for GPS time series before and after ICA/PCA filtering.

The remainder of this paper is organized as follows. Section 2 briefly reviews the theoretical background of the AIC and the stacking, PCA, and ICA techniques, data interpolation and factor analysis methods. Section 3 discusses the results of the PCA, ICA filters and the results of the optimal noise model. The comparison between the ICA-extracted CME and PCA-extracted CME and the noise results after applying PCA and ICA are presented in Section 4. The conclusion of our findings is presented at the end.

## 2. Materials and Methods

### 2.1. PCA

As presented by the authors of [23], daily stations within regional GNSS networks coordinate time series with $n$ stations and $m$ days. Accordingly, the matrix $X(m \times n)$ is constructed ($m > n$), where each column contains the detrended and demeaned coordinate values for a single geodetic component (north, east, vertical) from a single station in the network, and the rows contain geodetic component values for all stations at a given epoch. The covariance matrix $B$ used in our presentation is defined as:

$$b_{ij} = \frac{1}{m-1} \sum_{k=1}^{m} X(t_k, x_i) X(t_k, x_j). \tag{1}$$

$B(n \times n)$ is a symmetric matrix, and we decompose $B$ as:

$$B = V \Lambda V^T \tag{2}$$

where $\Lambda$ has $k$ nonzero diagonal eigenvalues (n >= k), $V^T$ is an (n × n) matrix with orthonormal rows, and we chose the orthonormal function basis $V$ to expand the data matrix $X$:

$$X(t_i, x_j) = \sum_{k=1}^{n} a_k(t_i) v_\kappa(x_j), \tag{3}$$

$$a_k(t_i) = \sum_{j=1}^{n} X(t_i, x_j) v_\kappa(x_j),$$ (4)

where $a_k(t)$ is the $k$th component of the matrix X representing the time variation and $v_k(x)$ is the spatial response of $a_k(t)$. If the eigenvalues are sorted in descending order, the first few PCs are the CME, whereas the higher-order PCs are related to local or individual site effects [23]. Hence, we can define the CME of PCA as follows:

$$CME_{PCA}(t_i, x_j) = \sum_{k=1}^{p} a_k(t_i) v_k(x_j),$$ (5)

where $p$ is the number of PCs used to compute CME.

*2.2. ICA*

As presented by the authors of [35], if we assume that there exist N underlying sources $s$, and X is the station time-series (collected from $n$ sites is a linear mixture of $s$), the instantaneous mixing model can be used, where any time delays that might occur in the mixing are neglected. At epoch $t$, $t = 1, 2,$ ..., $m$, we can write the mixing model as:

$$x_t = As_t + e_t,$$ (6)

where $x_t$, $s_t$, and $e_t$ are the observation vector, source signal, and systematic error or random noise at epoch $t$, respectively. **A** is the mixing matrix, and the element $a_{ij}$ is the contribution of the $j$-th source signal to the $i$-th observation ($1 \le i \le n, 1 \le j \le N$), which is also called the spatial response. Considering all epochs $m$, the matrix form of Equation (6) is:

$$X = AS + e,$$ (7)

with X = [$x1, x2, \ldots, xm$], S = [$s1, s2, \ldots, sm$], and e = [$e1, e2, \ldots, em$]. Because the statistic characteristic of $e$ is usually unknown, PCA is commonly used as a preprocessing step known as whitening. Then, Equation (7) becomes a noise-free model [46]. The goal of ICA is to find the unmixing matrix W that maximizes the non-Gaussianity of each source. The mixing matrix W is the inverse matrix of A and can be used to recover the original signal:

$$W = A^{-1}$$ (8)

Because the CME is a spatially correlated error, the ICA approach is used to extract the CME:

$$CME_{ICA} = \sum_{j \in R}^{R} \check{A}_j \overline{S_j}$$ (9)

where $\overline{S}$ is the ICA subset with common spatial characteristics and $\check{A}_j$ is the spatial response of $\overline{S_j}$.

*2.3. Akaike Information Criterion*

The qualities of the selected noise models in describing the noise in the data are evaluated using the AIC [44], which uses the log-likelihood as the starting point but adds penalties for adding parameters to avoid overfitting. The definition of the log-likelihood is as follows:

$$ln(L) = -\frac{1}{2}\left[Nln\ (2\pi) + lndet(C) + \mathbf{r}^T C^{-1}\right]$$ (10)

where $N$ is the actual number of observations (gaps do not count) and r is the residual vector of time series. The covariance matrix $C$ is decomposed as:

$$C = \sigma^2 \overline{C}, \tag{11}$$

where $\overline{C}$ is the sum of various noise models and σ is the standard deviation of the driving WN process, where σ is estimated from the residuals:

$$\sigma = \sqrt{\frac{r^T \overline{C}^{-1} r}{N}} \tag{12}$$

Then, the $AIC$ can be defined as follows:

$$IC = 2k + 2ln(L) \tag{13}$$

Because $det c A = c^N det A$, the following formulation for the likelihood is implemented:

$$ln(L) = -\frac{1}{2}\left[Nln(2\pi) + lndet\left(\overline{C}\right) + 2Nln(\sigma) + N\right]. \tag{14}$$

The number of parameters $k$ is the sum of the parameters in the design matrix and the noise models with the variance of the driving WN process. The preferred model is the one with the minimum AIC value.

### 2.4. Data Interpolation

The GPS time series are downloaded from the Nevada Geodetic Laboratory (http://geodesy.unr.edu/NGLStationPages/GlobalStationList). Based on the distribution and integrity of the GPS time series, we select 80 GPS stations in Antarctica with a time span from 2010.02.08 to 2018.06.23. Stations PAL2/PALV/PALM are located at the same site, as are stations ROBI/ROBN. Figure 1 shows the locations of the 80 GNSS stations in Antarctica.

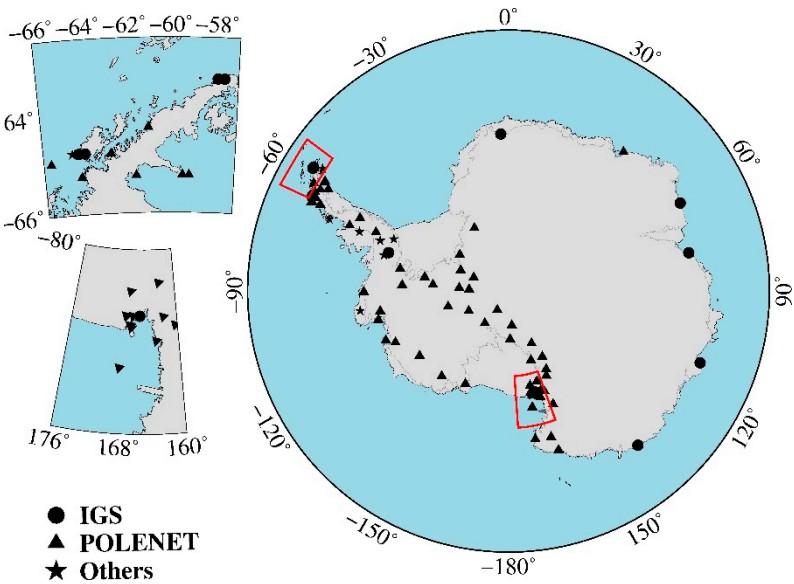

**Figure 1.** The distribution of Global Positioning System (GPS) stations in Antarctica.

We removed abnormal data from the raw time series with the third quartile criterion, which is based on the value of sigma to find and remove the data larger than the third quartile. For each coordinate time series, we estimated a constant offset and a trend in addition to annual and semiannual

terms. Then, we subtracted these terms from the coordinate time series to form the residual time series. In this work, before performing PCA and ICA, we used the regularized expectation-maximization (RegEM) algorithm to interpolate missing values [47,48]. RegEM, which neither depends on the data model nor introduces a priori information, relies on the self-characteristics of the data to compute the missing values while taking the physical background of the time series and the correlation among the GNSS time series into account. The average proportion of missing data in our time series is 25.54%. Figure 2 shows the results of RegEM interpolation for station CAPF; the black dots represent the raw residual time series, and the blue lines are the data interpolated by RegEM.

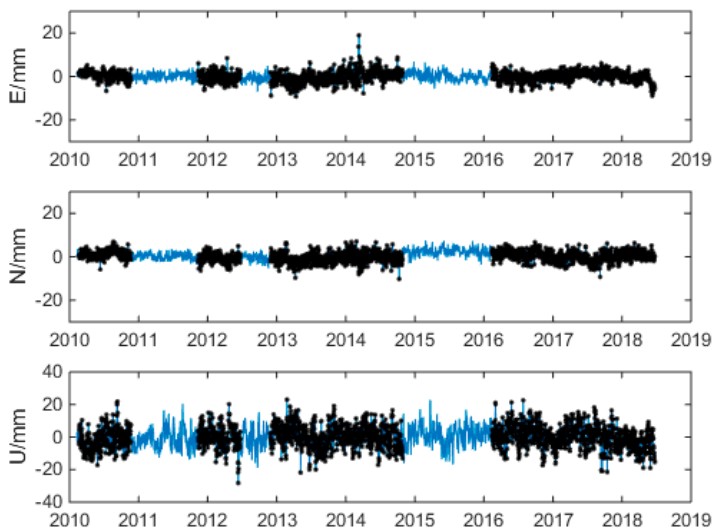

**Figure 2.** The RegEM interpolation results of CAPF stations. (The black dots represent the raw residual time series and the blue lines are the data interpolated by RegEM).

*2.5. Factor Analysis*

We first used the Kaiser–Meyer–Olkin (KMO) test [49,50] to validate the applicability of PCA and ICA for the GNSS data. The KMO test can provide measures of both the simple correlation coefficient and the partial correlation coefficient to validate the applicability of factor analysis for data. As the quadratic sum of the simple correlation coefficient and partial correlation coefficient grows larger, the KMO becomes closer to 1, indicating that the correlation is stronger, which means that the data can be used for factor analysis; otherwise, if the correlation is weak, the data cannot be used for factor analysis. The results showed that the KMO measures for the GNSS residual time series are 0.931, 0.941, and 0.963 in the east (E), north (N), and vertical (U) components, respectively. Furthermore, Bartlett's test can be used to test the statistical significance of the first PC [35,51]. Bartlett's test statistics are less than 0.01 for the data herein, indicating that the GNSS residual time series are correlated, and thus, the data can be used for factor analysis.

We perform singular value decomposition for the covariance of the E, N, and U residual time series; the eigenvalue spectra and percentage of the cumulative variance are displayed in Figures 3 and 4, respectively. From Figure 3, we can see that the eigenvalues of the E and N components are approximately equivalent, and they are less than that of the U component. Figure 4 shows that the first three eigenvalues collectively account for 46.96%, 51.68%, and 50.19% of the total variance of the E, N, and U components, respectively, and the first eigenvalues contribute 20.18%, 26.14%, and 28.01%, respectively, indicating that the signals on the E, N, and U components are described mainly by PC1. To confirm how many PCs are statistically significant, we perform parallel analysis (PA), a Monte Carlo-based simulation method that compares the observed eigenvalues with those obtained from simulated datasets. Here, a PC is retained if the associated eigenvalue is larger than 99% of the distribution of eigenvalues derived from random data [51]. Figure 5 is the PA test results of the east

(E) components (N, U are analogous), the lateral axis is the component number, and the vertical axis is the eigenvalues of principal components, from which we can see that the first 10 eigenvalues are statistically significant for the E, N, and U components. However, because the magnitudes of the PCs after PC3 are too small, their effects can be neglected, and thus, we analyze only the first three PCs.

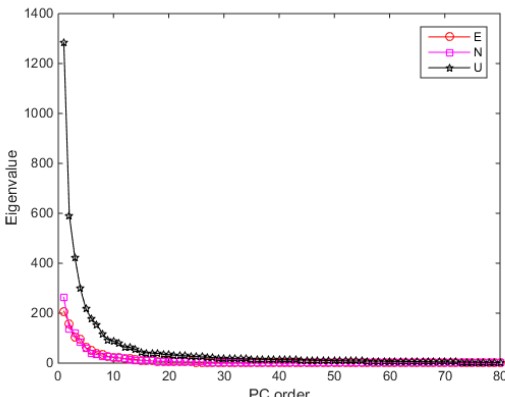

**Figure 3.** The eigenvalue spectrum of the east (E), north (N) and vertical (U) components derived from the PCA method.

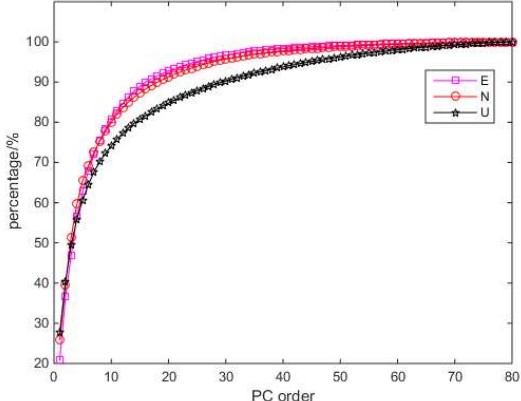

**Figure 4.** Percentage of cumulative variance.

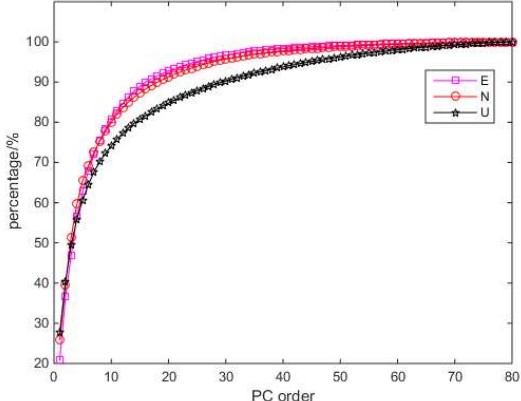

**Figure 5.** The parallel analysis (PA) test results of east (E) components.

## 3. Results

### *3.1. Reginal Filer Results*

#### 3.1.1. PCA Results

Based on the discussion provided in Section 2.1, the PCA filter is applied to the E, N, and U components, from which we find that station ROBI has a spatial response that is different from those of the other stations, while station ROBN (located almost at the same location as station ROBI) has a spatial response that is identical to those of the other stations. Hence, we suppose that station ROBI is abnormal and consequently remove it. Then, the remaining 79 stations are filtered by PCA. Figures 6–8 show the first three PCs for the E, N, U components, respectively. In these figures, upward arrows represent positive spatial responses (SRs) to the scaled PCs, while downward arrows represent negative SRs to the scaled PCs, where the legends represent a 100% SR.

Figure 6 shows the SRs of the first three PCs to the E component. Evidently, the spatial responses of PC1, PC2, and PC3 are neither completely random nor identical, but they exhibit obvious spatially uniform localized patterns or strong spatial coherence across the network. The SRs of PC1 are negative in the Ross Sea and East Antarctica and positive elsewhere, whereas the SRs of PC2 are negative in East Antarctica and Queen Maud Land and positive elsewhere, and the SRs of PC3 are negative on the Ronne Ice Shelf and in Dronning Maud Land and positive elsewhere.

Figure 7 shows the SRs of the first three PCs to the N component. The first two components also exhibit an obvious spatially uniform localized pattern. The SRs of PC1 are negative in the Ross Sea and East Antarctica, while those in the other regions are positive, whereas the SRs of PC2 are negative on the Ronne Ice Shelf and in Dronning Maud Land and positive elsewhere. By contrast, the SRs of PC3 exhibit spatially uniform localized patterns in some areas, but their patterns are not entirely uniform. Based on the results shown in Figure 7, we suppose that unmodeled signals, local effects, and noises among other factors not considered herein are present in the higher-order PCs.

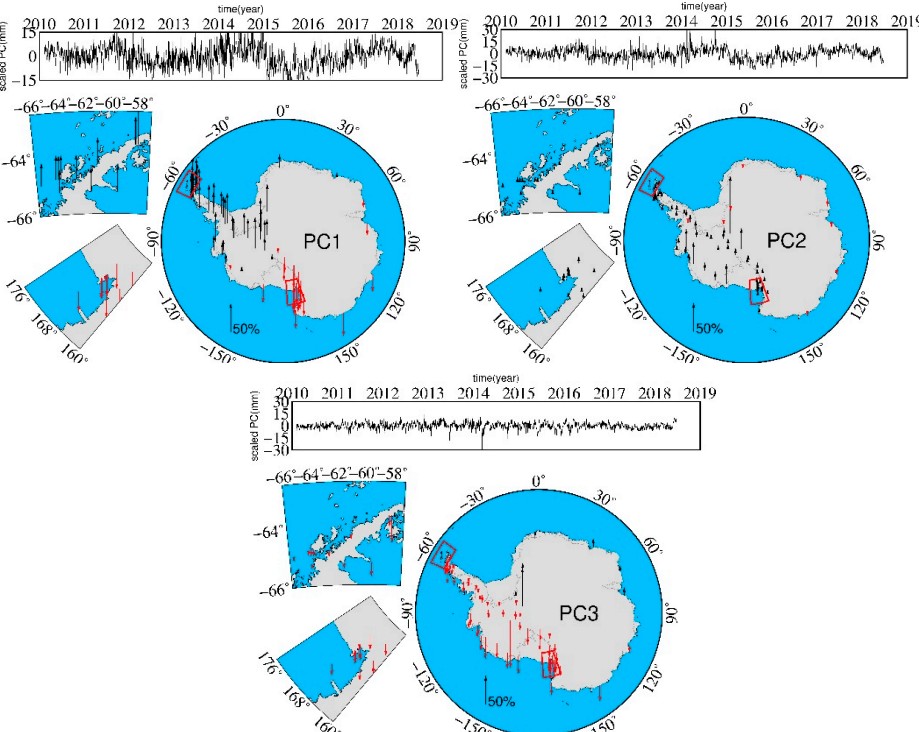

**Figure 6.** First three PCs for E components using PCA (the upward arrows are positive spatial response (SR), the downward are negative SR).

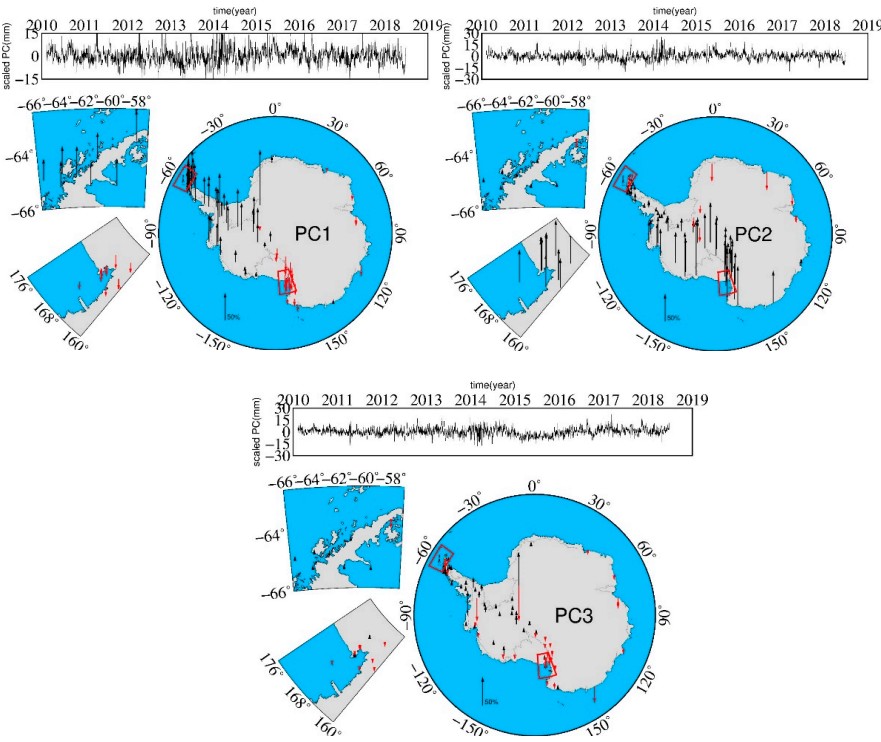

**Figure 7.** First three PCs for N components using the PCA (the upward arrows are positive SR, the downward are negative SR).

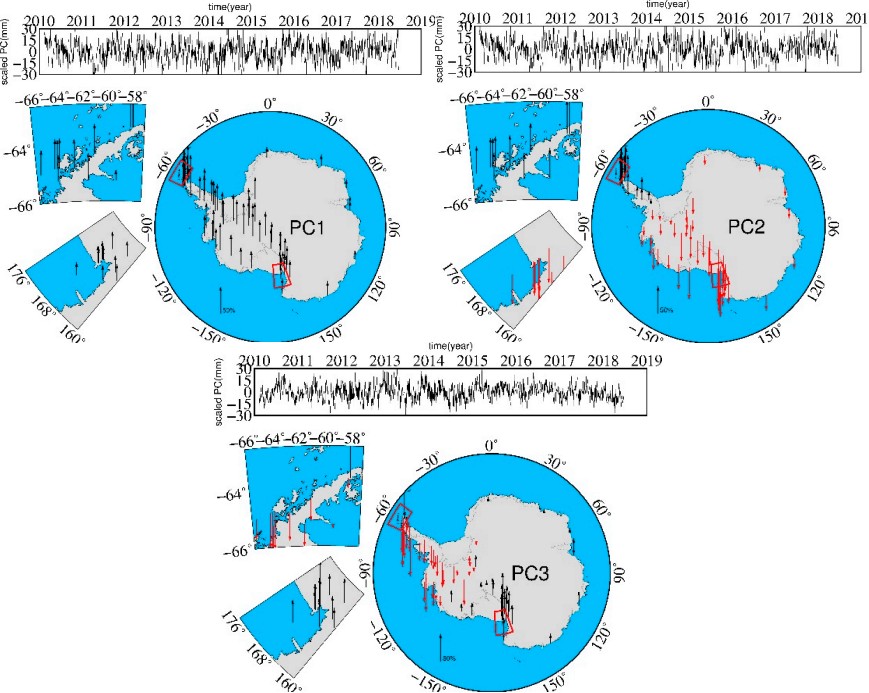

**Figure 8.** First three PCs for U components using the PCA method (the upward arrows are positive SR, the downward are negative SR).

Figure 8 shows the SRs of the first three PCs to the U component. The SRs of PC1 exhibit relatively uniform patterns, while the SRs of PC2 are positive in the Antarctica Peninsula (which may be affected by the collapse of the Larsen B Ice Shelf) and negative elsewhere. Meanwhile, the SRs of PC3 exhibit

similarly negative patterns in the Antarctica peninsula and in the regions around the Ronne Ice Shelf and Amundsen Coast.

The SRs of PC3 are influenced mainly by local effects based on the results shown in Figures 6 and 8, which demonstrate that the SRs of PC3 exhibit obvious regional spatial patterns in the E and U components; we suppose that these patterns are most likely attributable to the melting of ice and snow in West Antarctica. We treat the mode as the common mode if most sites (more than 50%) have significant normalized responses (larger than 25%) and if the eigenvalues of this mode exceed 1% of the summation of all eigenvalues [23]. According to this criterion, we choose the first three PCs of the E and U components and the first two PCs of the N component as the CME. For all stations excluded from the PCA because of long gaps in the data and strong local effects (e.g., the SRs of PC2 and PC3 in the E component at station PECE and in the E and U components at station OHI3), we use the average SRs for their CME corrections.

### 3.1.2. ICA Results

As with the abovementioned PCA, we again removed station ROBI and treated the residual time series as approximate whitening; then, we applied an ICA filter to the 79 GPS stations. Figures 9–11 show the results of the E, N, and U components, respectively, using the ICA method.

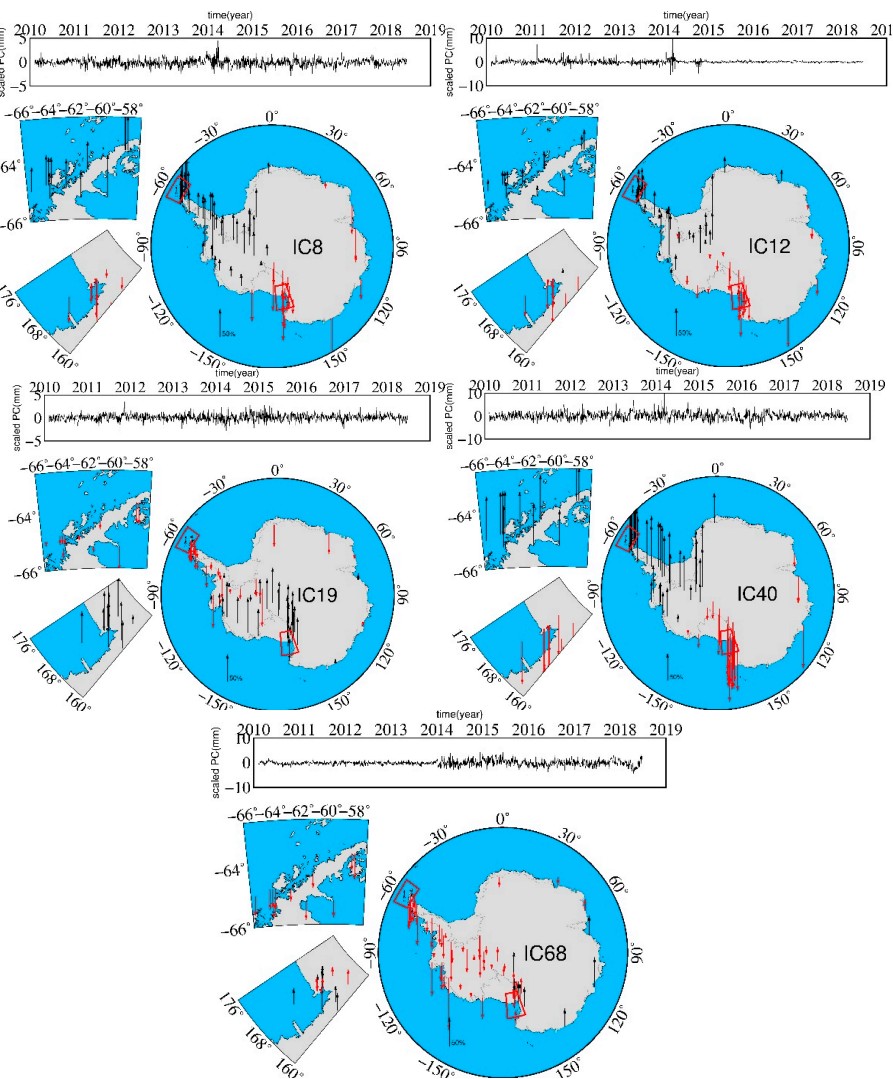

**Figure 9.** The results of east (E), IC8, IC12, IC19, IC40, and IC68 components using the ICA method (the upward arrows are a positive spatial response, the downward are a negative spatial response).

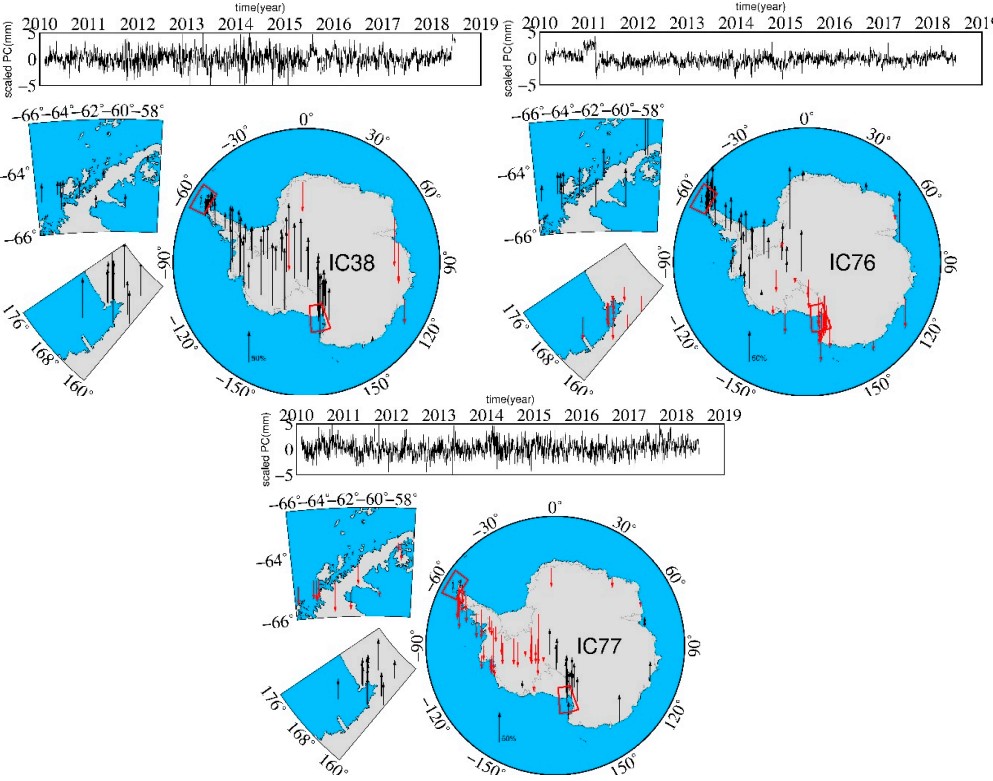

**Figure 10.** The results of north (N), IC38, IC76, and IC77 components using the ICA method (the upward arrows are a positive spatial response, the downward are a negative spatial response).

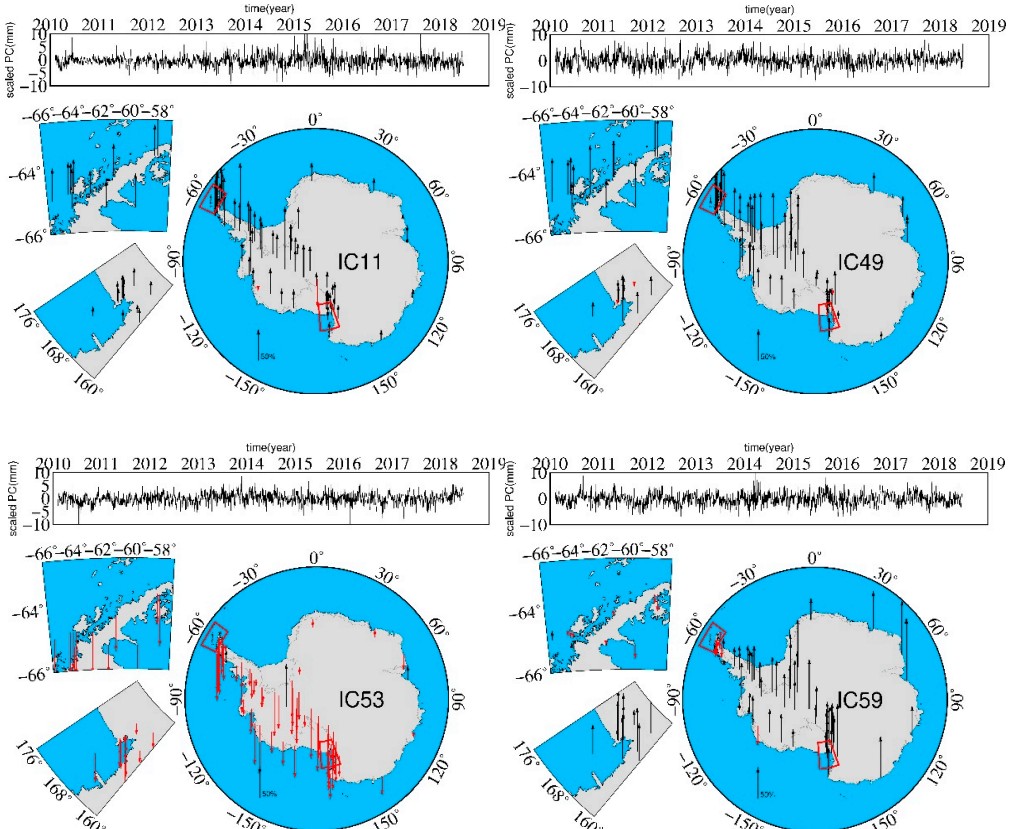

**Figure 11.** *Cont.*

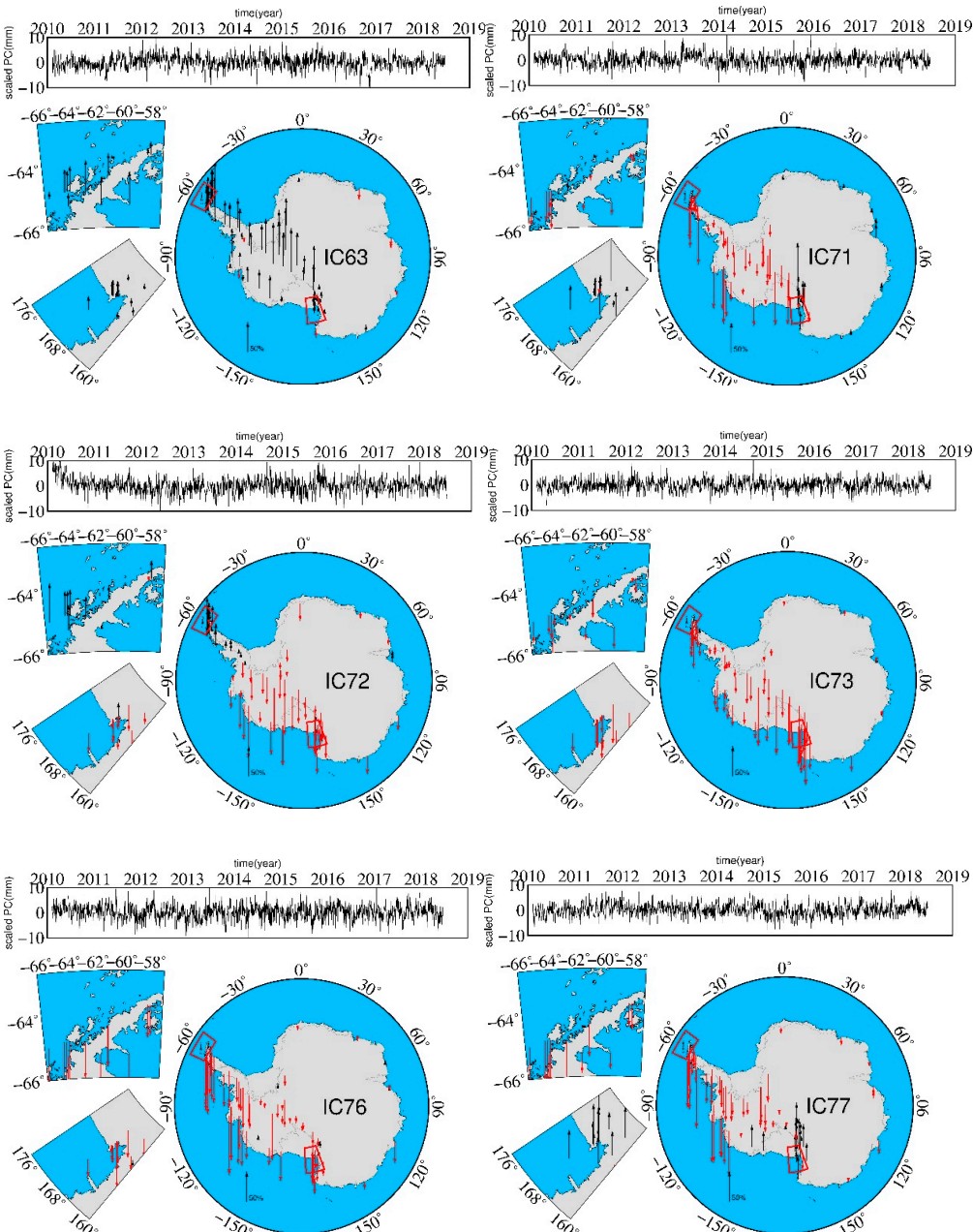

**Figure 11.** The results of vertical (U), IC11, IC49, IC53, IC59, IC63, IC71, IC72, IC73, IC76, and IC77 components using the ICA method (the upward arrows are a positive spatial response, the downward are a negative spatial response).

Calculations and subsequent analysis reveal that IC8, IC12, IC19, IC40, and IC68 of the E component, IC38, IC76, and IC77 of the N component and IC11, IC49, IC53, IC59, IC63, IC71, IC72, IC73, IC76, and IC77 of the U component have almost uniform spatial patterns, while the SRs of the other ICs have no obvious spatially uniform localized pattern or strong spatial coherence. The SRs of these ICs with uniform spatial patterns represent the corresponding uniform spatial patterns of the CME in the E, N, and U components. Based on these results, we calculate the CME for the E, N, and U components.

Regarding the E component, Figure 9 shows that the SRs of IC8 and IC40 are similar to those of PC1 derived from the PCA filter, but their magnitudes are larger than that of PC1. We infer that PC1 contains a mixture of multisource error signals and that these signals may cancel one another out. Analogously, from Figures 10 and 11, we find that the SRs of IC76 and IC38 are similar to those of PC1

and PC2, respectively, in the N component; in addition, the SRs of IC11, IC49, and IC59 are similar to that of PC1 in the U component, and the SR of IC72 is similar to that of PC2. Compared with that derived from PCA, the CME derived from ICA exhibits more obvious spatial diversity and spatially uniform localized characteristics, which means that ICA can extract multisource error signals both effectively and successfully. However, the responses of some stations with small SRs revealed by the PCA and ICA filters contrast with those of nearby stations; we infer that these differences are caused by spatial noise derived from PCA- and ICA-modeled errors or by local effects.

Figure 12 shows the spectral analysis results of the temporal variations in the E, N, and U components derived from the ICA results. We find that low-frequency signals are evident in all three components. It is worth noting that the time series are of insufficient length to detect the lowest frequencies of IC40 in the E component and of IC76 in the U component. Based on the spectral analysis results, we extracted the periods of the first 11 peaks for ICs used to compute CME in the E, N, U components, which are listed in Table 1. The results show that seven signals with periods of 250, 214.3, 142.9, 115.4, 111.1, 76.9, and 52.6 days are shown in all three components. Nevertheless, the geophysical mechanism underlying this phenomenon requires further investigation.

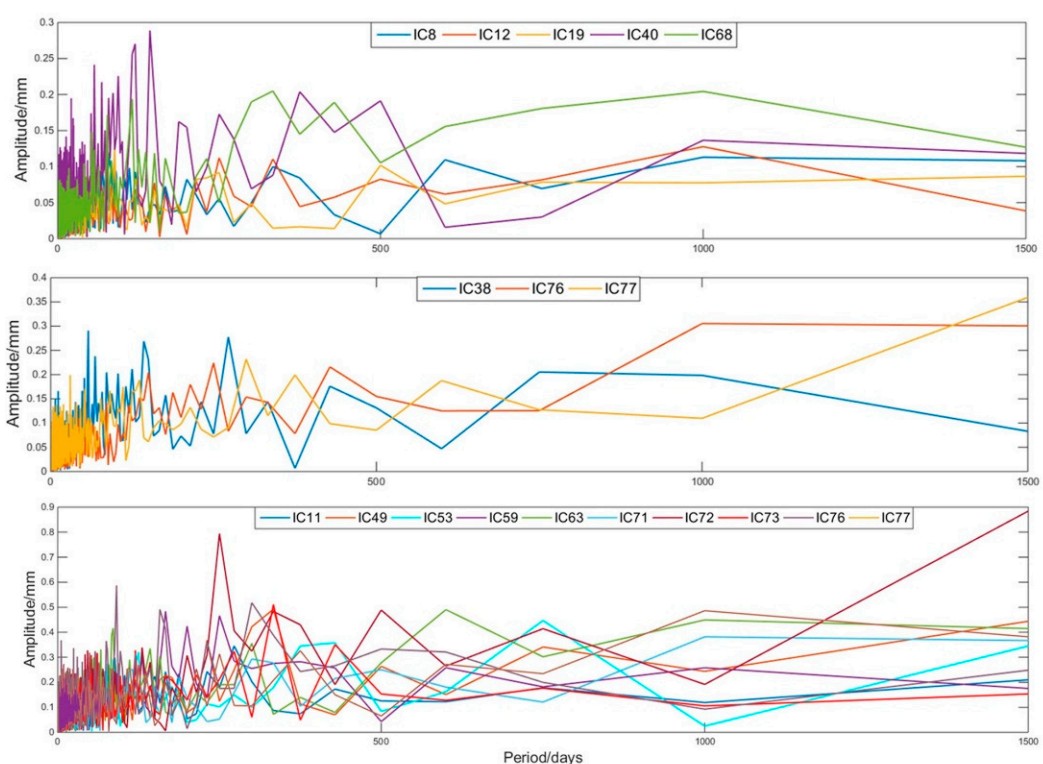

**Figure 12.** Results of spectral analysis of temporal variations: east (**top**), north (**middle**), and vertical (**bottom**).

We also extracted the CME derived from the ICA results with the criterion employed in the PCA, that is, we treated the mode as the common mode if most sites (more than 50%) have significant normalized responses (larger than 25%) and if the eigenvalues of this mode exceed 1% of the summation of all eigenvalues [23]. According to this criterion, the following ICs were chosen to compute the CME using the ICA method: IC8, IC12, IC19, IC40, and IC68 for the E component, IC38, IC76, and IC77 for the N component, and IC11, IC49, IC53, IC59, IC63, IC71, IC72, IC73, IC76, and IC77 for the U component.

**Table 1.** The periods of the first 11 peaks for ICs used to computed CME in the E, N, U components (days).

|  | **1** | **2** | **3** | **4** | **5** | **6** | **7** | **8** | **9** | **10** |
|---|---|---|---|---|---|---|---|---|---|---|
| **E** | | | | | | | | | | |
| IC8 | 78.9 | 1000.0 | 600.0 | 333.3 | 88.2 | **111.1** | 69.8 | 120.0 | 103.4 | 200.0 |
| IC12 | 1000 | **250.0** | 333.3 | 500.0 | **214.3** | 150.0 | 12.4 | 83.3 | 68.2 | 14.3 |
| IC19 | 88.2 | 500.0 | **250.0** | 1500.0 | 48.4 | 120.0 | 40.5 | 750.0 | 150.0 | 61.2 |
| IC40 | **142.9** | 120.0 | 56.6 | 93.8 | 68.2 | 375.0 | 85.7 | 78.9 | 21.0 | 500.0 |
| IC68 | 333.3 | 1000.0 | **115.4** | 428.6 | **76.9** | **52.6** | 125.0 | 56.6 | 136.4 | 150.0 |
| **N** | | | | | | | | | | |
| IC38 | 57.7 | 272.7 | **142.9** | 68.2 | 125.0 | 750.0 | 85.7 | 103.4 | 51.7 | **115.4** |
| IC76 | 3000 | 1000.0 | **250.0** | 428.6 | 150.0 | **214.3** | 187.5 | 300.0 | 136.4 | 53.6 |
| IC77 | 1500 | 300.0 | 375.0 | 30.0 | 136.4 | 600.0 | **111.1** | **52.6** | 90.9 | **76.9** |
| **U** | | | | | | | | | | |
| IC11 | 272.7 | 88.2 | 230.8 | 166.7 | 13.2 | 5.4 | 187.5 | 78.9 | 103.4 | 24.0 |
| IC49 | 333.3 | 1500.0 | 230.8 | 750.0 | 93.8 | 500.0 | 63.8 | 81.1 | 75.0 | 66.7 |
| IC53 | 750 | 428.6 | 3000.0 | 125.0 | 85.7 | **76.9** | 38.0 | 69.8 | 107.1 | 43.5 |
| IC59 | 166.7 | **250.0** | 200.0 | 130.4 | 375.0 | 600.0 | 1000.0 | 6.3 | 90.9 | 85.7 |
| IC63 | 3000 | 600.0 | 1000.0 | 85.7 | 300.0 | **142.9** | 66.7 | 157.9 | 55.6 | **52.6** |
| IC71 | 1000 | 300.0 | 63.8 | **115.4** | 500.0 | 71.4 | 5.5 | 5.4 | 187.5 | 60.0 |
| IC72 | 1500 | **250.0** | 500.0 | 333.3 | 750.0 | 130.4 | 46.2 | 42.3 | 51.7 | 200.0 |
| IC73 | 333.3 | 428.6 | 120.0 | 272.7 | 61.2 | 51.7 | **214.3** | 96.8 | 166.7 | 5.3 |
| IC76 | 90.9 | 300.0 | 157.9 | 3000.0 | 5.3 | 75.0 | 230.8 | 24.6 | 500.0 | 96.8 |
| IC77 | 1000 | 3000.0 | 375.0 | 49.2 | **250.0** | 600.0 | 125.0 | 8.6 | 111.1 | 83.3 |

*3.2. Noise Analysis*

The noise model is one of the most important factors affecting the precision of GNSS coordinate time series. Previous researchers have shown that GNSS time series contain not only WN but also FN and RW. In GNSS time series, the velocity uncertainty is usually influenced prominently by ignoring the effects of colored noise, which leads to incorrect physical interpretations. For regions with a vast spatial area and complex terrain such as Antarctica, the system errors, random errors and local effects as well as the optimal noise model will be quite different among the stations in a network. Consequently, the use of only one noise model is insufficient to reasonably and effectively model all GNSS station time series. To determine the optimal noise model for Antarctica, we use a combination of 5 noise models supplied by Hector [44] to analyze the 79 GNSS station time series: WN+PN, WN+RW, WN+FN, WN+FN+RW, and WN+RW+GGM. Then, we determine the optimal noise model for all stations based on the AIC. Figures 13–15 show the percentage of stations optimal noise model for the time series before and after using the PCA and the ICA filter in the E, N, and U components, respectively; we found that there are more stations with consistent optimal noise models after ICA filtering than there are after PCA filtering. The numbers of stations after applying the ICA filter were 5, 69, and 63 in the E, N, and U components, respectively, while the numbers of stations after applying the PCA filter are 41, 49, and 62, respectively. Table 2 lists the statistic of sites optimal models. From Figures 13 and 14 and Table 2, we can conclude that the optimal noise models of the raw residual time series are mainly WN+PN, WN+FN, and WN+RW+GGM. After applying the PCA and ICA filters, the number of stations for which the WN+FN model is the optimal model reduces, and the number of stations for which the WN+RW+GGM model is the optimal model increases, indicating that the PCA and ICA filters may change the model characteristics of some station time series. Figure 15 and Table 2 show that the optimal model (the WN+PN model) for the raw residual series and for the time series after applying the PCA and ICA filters in the U component accounts for 61%, 70%, and 75% of all stations, respectively. After applying the PCA and ICA filters, the number of stations for which the WN+PN model is the optimal model increases, while that for which the WN+FN model is the optimal model decreases. Furthermore, we calculate the PN spectral index and find that most of the

PN spectral index approximates the FN, which indicates that the essence of PN is similar to that of FN in Antarctica.

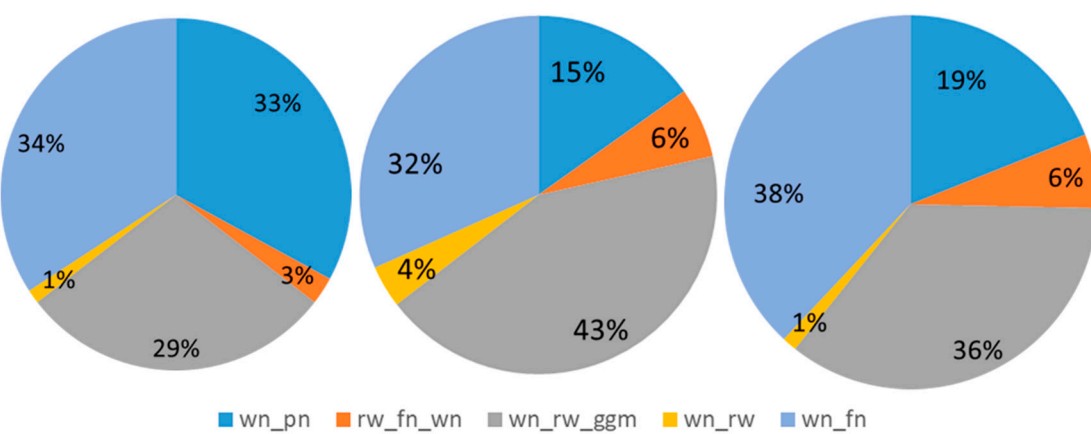

**Figure 13.** The optimal noise model results of time series before (**left**) and after the PCA (**middle**), ICA (**right**) filtering in the E components.

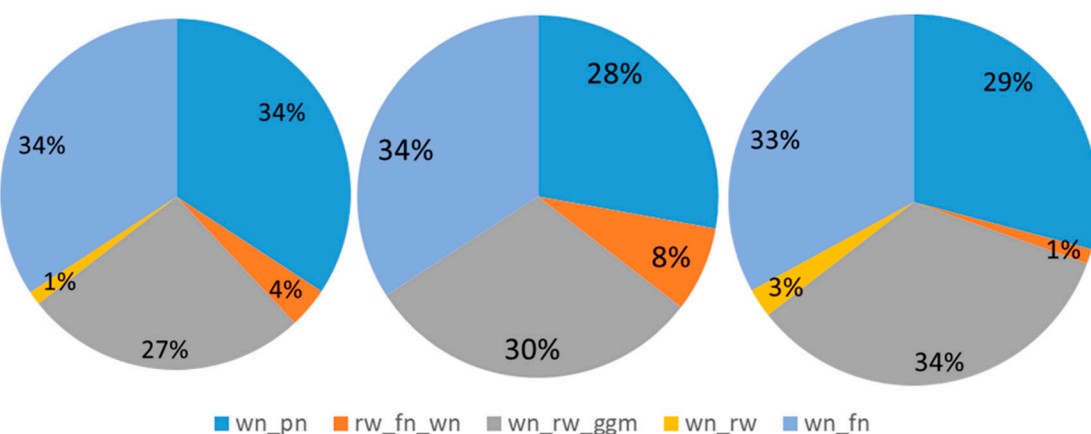

**Figure 14.** The optimal noise model results of time series before (**left**) and after the PCA (**middle**), ICA (**right**) filtering in the N components.

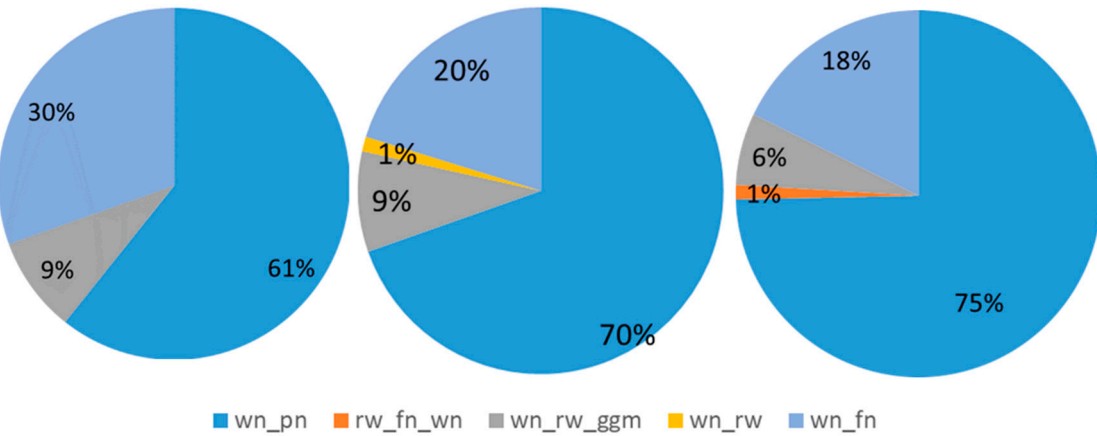

**Figure 15.** The optimal noise model results of time series before (**left**) and after the PCA (**middle**), ICA (**right**) filter in the U components.

**Table 2.** The statistic of sites optimal models.

| Sites | RAW | | | PCA | | | ICA | | |
|---|---|---|---|---|---|---|---|---|---|
| | **E** | **N** | **U** | **E** | **N** | **U** | **E** | **N** | **U** |
| **ABBZ** | WN+FN | WN+PN | WN+PN | WN+FN | WN+PN | WN+PN | WN+FN | WN+PN | WN+PN |
| **BACK** | WN+PN | WN+FN | WN+FN | WN+RW+GGM | WN+FN | WN+PN | WN+FN | WN+FN | WN+PN |
| **BENN** | WN+FN | WN+RW+GGM | WN+PN | WN+RW+GGM | WN+RW+GGM | WN+PN | WN+FN | WN+RW+GGM | WN+PN |
| **BERP** | WN+RW+GGM | WN+FN | WN+PN | WN+RW+GGM | WN+FN | WN+PN | WN+RW+FN | WN+FN | WN+PN |
| **BRIP** | WN+PN | WN+PN | WN+PN | WN+PN | WN+PN | WN+PN | WN+PN | WN+PN | WN+PN |
| **BUMS** | WN+PN | WN+FN | WN+FN | WN+FN | WN+FN | WN+FN | WN+FN | WN+PN | WN+FN |
| **BURI** | WN+PN | WN+PN | WN+PN | WN+FN | WN+FN | WN+PN | WN+FN | WN+PN | WN+PN |
| **CAPF** | WN+FN | WN+FN | WN+FN | WN+PN | WN+FN | WN+PN | WN+RW+GGM | WN+FN | WN+PN |
| **CAS1** | WN+FN | WN+FN | WN+PN | WN+FN | WN+FN | WN+PN | WN+FN | WN+FN | WN+PN |
| **CLRK** | WN+FN | WN+FN | WN+PN | WN+RW+GGM | WN+FN | WN+PN | WN+RW+GGM | WN+FN | WN+PN |
| **COTE** | WN+PN | WN+PN | WN+PN | WN+FN | WN+PN | WN+PN | WN+FN | WN+PN | WN+PN |
| **CRAR** | WN+PN | WN+PN | WN+RW+GGM | WN+PN | WN+FN | WN+RW+GGM | WN+PN | WN+PN | WN+RW+GGM |
| **CRDI** | WN+RW+GGM | WN+RW+GGM | WN+PN | WN+RW+GGM | WN+RW+GGM | WN+PN | WN+RW+GGM | WN+RW+GGM | WN+PN |
| **DAV1** | WN+FN | WN+FN | WN+PN | WN+FN | WN+FN | WN+PN | WN+PN | WN+FN | WN+PN |
| **DAVE** | WN+FN | WN+FN | WN+RW+GGM | WN+FN | WN+FN | WN+RW+GGM | WN+PN | WN+FN | WN+RW+GGM |
| **DEVI** | WN+PN | WN+PN | WN+PN | WN+RW+GGM | WN+PN | WN+PN | WN+FN | WN+PN | WN+PN |
| **DUM1** | WN+RW+GGM | WN+RW+GGM | WN+FN | WN+RW+GGM | WN+RW+GGM | WN+FN | WN+RW+GGM | WN+RW+GGM | WN+FN |
| **DUPT** | WN+FN | WN+FN | WN+PN | WN+FN | WN+PN | WN+PN | WN+RW+GGM | WN+FN | WN+PN |
| **FALL** | WN+FN | WN+PN | WN+PN | WN+RW+GGM | WN+FN | WN+PN | WN+FN | WN+PN | WN+PN |
| **FIE0** | WN+PN | WN+PN | WN+PN | WN+FN | WN+FN | WN+PN | WN+PN | WN+PN | WN+PN |
| **FLM5** | WN+PN | WN+PN | WN+PN | WN+PN | WN+PN | WN+PN | WN+PN | WN+PN | WN+PN |
| **FONP** | WN+FN | WN+FN | WN+FN | WN+FN | WN+FN | WN+PN | WN+FN | WN+FN | WN+PN |
| **FOS1** | WN+FN | WN+FN | WN+PN | WN+FN | WN+PN | WN+PN | WN+FN | WN+FN | WN+PN |
| **FTP4** | WN+PN | WN+PN | WN+PN | WN+FN | WN+FN | WN+PN | WN+PN | WN+PN | WN+PN |
| **GMEZ** | WN+FN | WN+RW+GGM | WN+FN | WN+RW+GGM | WN+RW+GGM | WN+FN | WN+RW+GGM | WN+RW+GGM | WN+PN |
| **HAAG** | WN+RW+GGM | WN+RW+GGM | WN+FN | WN+RW+GGM | WN+RW+GGM | WN+PN | WN+RW+GGM | WN+RW+GGM | WN+PN |
| **HOOZ** | WN+PN | WN+PN | WN+PN | WN+RW+GGM | WN+PN | WN+PN | WN+FN | WN+PN | WN+PN |
| **HOWE** | WN+RW+GGM | WN+RW+GGM | WN+FN | WN+RW+FN | WN+RW+GGM | WN+FN | WN+RW+GGM | WN+RW+GGM | WN+FN |
| **HOWN** | WN+RW+GGM | WN+RW+GGM | WN+PN | WN+RW+GGM | WN+RW+GGM | WN+PN | WN+RW+GGM | WN+RW+GGM | WN+PN |
| **HUGO** | WN+FN | WN+PN | WN+FN | WN+FN | WN+FN | WN+RW+GGM | WN+FN | WN+FN | WN+PN |
| **IGGY** | WN+RW+GGM | WN+RW+GGM | WN+FN | WN+RW | WN+RW+GGM | WN+FN | WN+RW+GGM | WN+RW+GGM | WN+FN |
| **INMN** | WN+RW+GGM | WN+RW+FN | WN+FN | WN+RW+FN | WN+RW+GGM | WN+FN | WN+RW+GGM | WN+RW+GGM | WN+FN |
| **JNSN** | WN+FN | WN+RW+GGM | WN+FN | WN+RW+GGM | WN+RW+GGM | WN+FN | WN+FN | WN+RW+GGM | WN+FN |
| **LNTK** | WN+RW+GGM | WN+RW+GGM | WN+FN | WN+RW+GGM | WN+RW+FN | WN+FN | WN+RW+GGM | WN+RW+GGM | WN+FN |
| **LPLY** | WN+RW+GGM | WN+RW+GGM | WN+FN | WN+RW+FN | WN+RW+FN | WN+FN | WN+RW+GGM | WN+RW+GGM | WN+FN |
| **LWN0** | WN+PN | WN+PN | WN+PN | WN+FN | WN+FN | WN+PN | WN+FN | WN+PN | WN+PN |
| **MACG** | WN+FN | WN+PN | WN+PN | WN+RW+GGM | WN+PN | WN+FN | WN+RW+GGM | WN+PN | WN+PN |
| **MAW1** | WN+FN | WN+PN | WN+PN | WN+FN | WN+PN | WN+PN | WN+FN | WN+FN | WN+PN |
| **MBIO** | WN+FN | WN+RW+GGM | WN+PN | WN+RW+GGM | WN+RW+FN | WN+PN | WN+RW+GGM | WN+RW+GGM | WN+PN |

**Table 2.** *Cont.*

| Sites | RAW | | | PCA | | | ICA | | |
|---|---|---|---|---|---|---|---|---|---|
| | **E** | **N** | **U** | **E** | **N** | **U** | **E** | **N** | **U** |
| MCAR | WN+RW+GGM | WN+FN | WN+PN | WN+RW+GGM | WN+RW+GGM | WN+PN | WN+RW+GGM | WN+RW+GGM | WN+PN |
| MCM4 | WN+PN | WN+PN | WN+PN | WN+PN | WN+FN | WN+PN | WN+FN | WN+PN | WN+PN |
| MCMD | WN+PN | WN+PN | WN+PN | WN+FN | WN+FN | WN+PN | WN+FN | WN+PN | WN+PN |
| MIN0 | WN+PN | WN+FN | WN+PN | WN+FN | WN+FN | WN+PN | WN+FN | WN+FN | WN+PN |
| MKIB | WN+RW+FN | WN+RW+GGM | WN+PN | WN+RW | WN+RW+GGM | WN+PN | WN+RW+FN | WN+RW+GGM | WN+PN |
| OHI2 | WN+RW+GGM | WN+PN | WN+RW+GGM | WN+RW+GGM | WN+FN | WN+PN | WN+RW+GGM | WN+FN | WN+FN |
| OHI3 | WN+FN | WN+FN | WN+PN | WN+FN | WN+FN | WN+PN | WN+FN | WN+FN | WN+PN |
| PAL2 | WN+FN | WN+PN | WN+FN | WN+PN | WN+PN | WN+PN | WN+PN | WN+FN | WN+PN |
| PALM | WN+PN | WN+PN | WN+FN | WN+PN | WN+PN | WN+PN | WN+PN | WN+FN | WN+PN |
| PALV | WN+PN | WN+FN | WN+PN | WN+FN | WN+PN | WN+PN | WN+FN | WN+FN | WN+PN |
| PATN | WN+RW+FN | WN+RW+GGM | WN+PN | WN+RW+FN | WN+RW+GGM | WN+PN | PATN+RW+FN | WN+RW+GGM | WN+PN |
| PECE | WN+RW | WN+RW+FN | WN+RW+GGM | WN+RW+GGM | WN+RW+FN | WN+RW | WN+RW | WN+RW | WN+RW+GGM |
| PHIG | WN+FN | WN+FN | WN+FN | WN+RW+GGM | WN+RW+GGM | WN+RW+GGM | WN+RW+GGM | WN+FN | WN+PN |
| PIRT | WN+PN | WN+FN | WN+PN | WN+PN | WN+PN | WN+PN | WN+PN | WN+RW+GGM | WN+PN |
| PRPT | WN+PN | WN+FN | WN+FN | WN+PN | WN+PN | WN+PN | WN+PN | WN+FN | WN+PN |
| RAMG | WN+PN | WN+PN | WN+PN | WN+FN | WN+PN | WN+PN | WN+FN | WN+PN | WN+PN |
| RMBO | WN+FN | WN+RW+GGM | WN+PN | WN+RW+GGM | WN+FN | WN+PN | WN+FN | WN+RW+GGM | WN+PN |
| ROB4 | WN+PN | WN+PN | WN+PN | WN+FN | WN+FN | WN+PN | WN+FN | WN+PN | WN+PN |
| ROBN | WN+FN | WN+FN | WN+FN | WN+FN | WN+RW+GGM | WN+PN | WN+RW+GGM | WN+RW+GGM | WN+PN |
| ROTH | WN+RW+GGM | WN+PN | WN+FN | WN+RW+GGM | WN+RW+GGM | WN+PN | WN+RW+GGM | WN+FN | WN+PN |
| SCTB | WN+FN | WN+FN | WN+PN | WN+RW+GGM | WN+FN | WN+FN | WN+FN | WN+FN | WN+PN |
| SDLY | WN+PN | WN+PN | WN+PN | WN+PN | WN+PN | WN+PN | WN+PN | WN+PN | WN+PN |
| SPGT | WN+FN | WN+FN | WN+FN | WN+FN | WN+PN | WN+PN | WN+FN | WN+FN | WN+PN |
| STEW | WN+RW+GGM | WN+FN | WN+PN | WN+RW+GGM | WN+FN | WN+PN | WN+RW+FN | WN+FN | WN+FN |
| SUGG | WN+RW+GGM | WN+RW+GGM | WN+PN | WN+RW+GGM | WN+RW+GGM | WN+FN | WN+RW+GGM | WN+RW+GGM | WN+FN |
| SYOG | WN+FN | WN+PN | WN+PN | WN+RW+GGM | WN+PN | WN+PN | WN+FN | WN+PN | WN+PN |
| THU4 | WN+RW+GGM | WN+RW+GGM | WN+PN | WN+RW+FN | WN+RW+FN | WN+PN | WN+RW+FN | WN+RW+GGM | WN+PN |
| TOMO | WN+RW+GGM | WN+RW+GGM | WN+RW+GGM | WN+RW+GGM | WN+RW+GGM | WN+RW+GGM | WN+RW+GGM | WN+RW+GGM | WN+RW+GGM |
| TRVE | WN+RW+GGM | WN+FN | WN+PN | WN+RW+GGM | WN+RW+GGM | WN+PN | WN+RW+GGM | WN+RW+GGM | WN+PN |
| VESL | WN+PN | WN+PN | WN+PN | WN+PN | WN+PN | WN+PN | WN+PN | WN+PN | WN+PN |
| VL01 | WN+PN | WN+PN | WN+PN | WN+PN | WN+PN | WN+PN | WN+PN | WN+PN | WN+PN |
| VL12 | WN+FN | WN+RW+GGM | WN+PN | WN+RW+GGM | WN+RW+GGM | WN+PN | WN+FN | WN+RW+GGM | WN+PN |
| VL30 | WN+RW+GGM | WN+RW+GGM | WN+FN | WN+RW+GGM | WN+RW+GGM | WN+FN | WN+RW+GGM | WN+RW+GGM | WN+FN |
| VNAD | WN+PN | WN+FN | WN+FN | WN+FN | WN+FN | WN+PN | WN+PN | WN+FN | WN+PN |
| WHN0 | WN+PN | WN+PN | WN+PN | WN+FN | WN+PN | WN+PN | WN+FN | WN+PN | WN+PN |
| WHTM | WN+RW+GGM | WN+RW | WN+RW+GGM | WN+RW+GGM | WN+RW+FN | WN+RW+GGM | WN+RW+GGM | WN+RW | WN+RW+FN |
| WILN | WN+RW+GGM | WN+RW+FN | WN+FN | WN+RW | WN+RW+GGM | WN+FN | WN+RW+GGM | WN+RW+FN | WN+FN |
| WLCH | WN+RW+GGM | WN+FN | WN+FN | WN+RW+GGM | WN+RW+GGM | WN+FN | WN+RW+GGM | WN+FN | WN+FN |
| WLCT | WN+FN | WN+FN | WN+PN | WN+RW+GGM | WN+FN | WN+PN | WN+FN | WN+RW+GGM | WN+PN |
| WWAY | WN+RW+GGM | WN+RW+GGM | WN+RW+GGM | WN+RW+GGM | WN+RW+GGM | WN+RW+GGM | WN+RW+GGM | WN+RW+GGM | WN+RW+GGM |

## 4. Discussion

### 4.1. Comparison Between the ICA-extracted CME and PCA-extracted CME

Figure 16 reveals the residual time series of station COTE before and after applying the regional filters using PCA (top) and ICA (bottom); the gray lines represent raw time series and the black lines are filtered results. The time series of daily east and north positions before and after filtering are shifted by an offset, for clarity. Clearly, the scattering in the filtered time series is reduced effectively by the PCA and ICA filters, and the reduction in the magnitude with the PCA filter is larger than that with the ICA filter (although the other stations have the same results) because the PCA-extracted CME is based on the maximum variance of the residual time series; thus, the second-order statistics of the PCA-filtered residual series are less than those of the ICA-filtered residual series. Moreover, PCA cannot take full advantage of higher-order statistics. During the application of the ICA filter, we use 5, 3, and 10 independent components to calculate the CME for the E, N, and U components, respectively, thereby introducing higher-order statistics and extracting multisource error signals effectively.

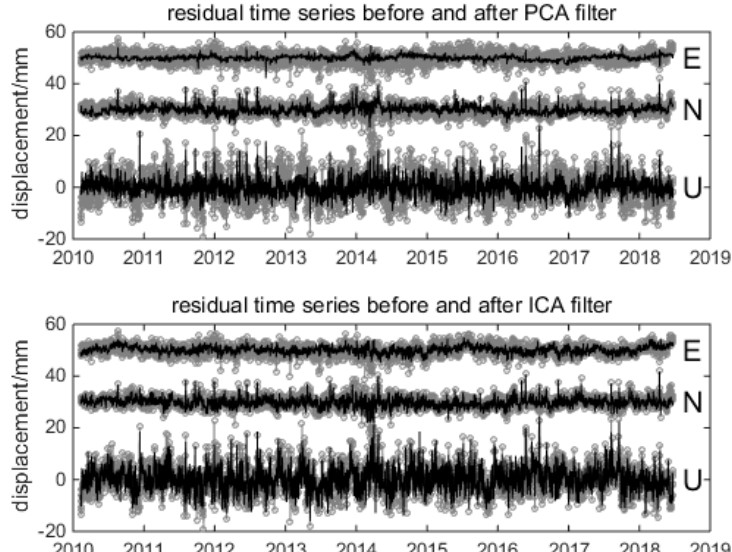

**Figure 16.** The residual time series of COTE stations before and after the regional filter using the PCA and ICA.

Figure 17 shows the root mean square (RMS) values of the residual time series before and after applying the PCA and ICA filters. The color bar represents the percentage of the RMS reduction, and the maximum, minimum and mean reductions in the RMS are listed in Table 3. It is worth noting that in Figure 17, the SRs of the CME derived from PCA have larger differences than those of the CME derived from ICA at 79 stations, and the ICA-extracted CME exhibits more obvious spatially uniform localized patterns, indicating that the ICA-extracted CME performs better in Antarctica. Table 3 shows that the mean reductions in the RMS values are 35.24%, 23.95%, and 30.41% in the E, N, and U components, respectively, after applying the PCA filter, and the corresponding mean reductions in the RMS values are 14.45%, 8.97%, and 13.27% after applying the ICA filter. It is worth noting that the PCA-based reduction in the RMS in the U component at station OHI3 is 81.96%, whereas the same reduction at station OHI2, which is sited at the same location, is only 27.40%. In addition, the reductions in the RMS values are only 7.05% and 1.32% at stations OHI2 and OHI3, respectively, after applying the ICA filter. Because PCA decomposition is based on second-order statistics, there is a risk of overfiltering with PCA [35]. We therefore believe that the PCA filter may remove the original signals in the U component at station OHI3 or that there are some unmodeled errors that PCA cannot remove; hence, we suppose that the ICA filter performs better than the PCA filter in Antarctica.

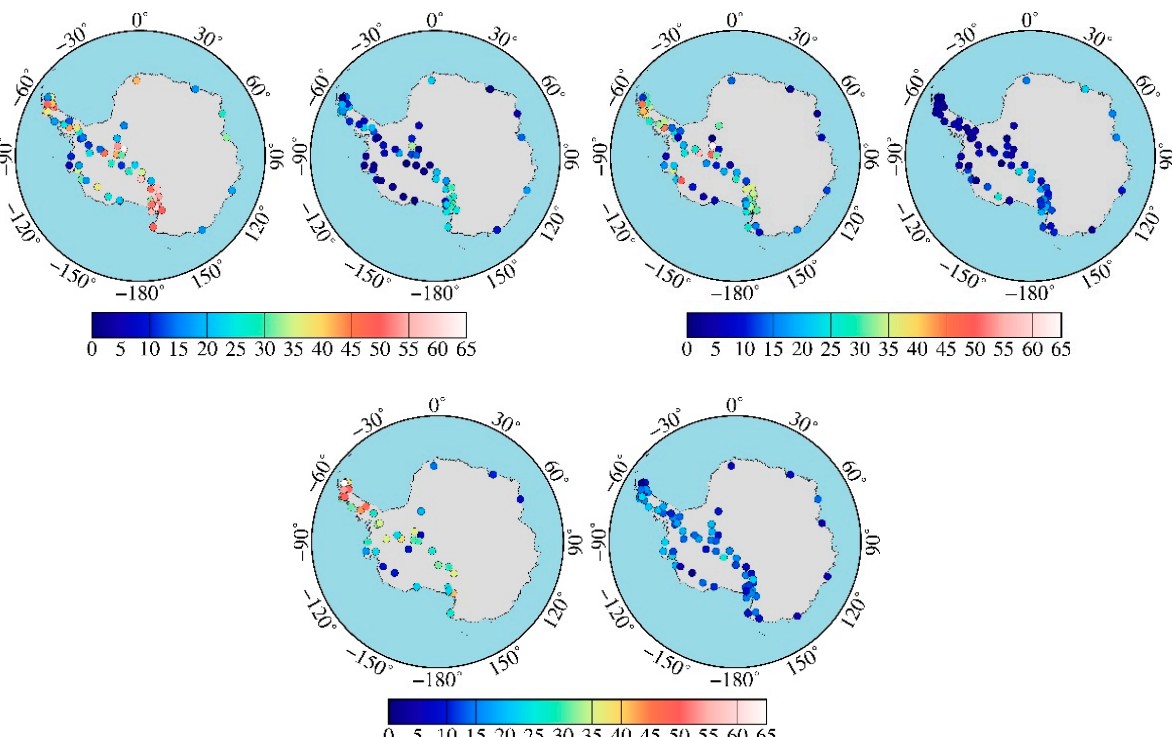

**Figure 17.** The root mean square(RMS) of residual time series before and after the PCA and ICA filtering (%); the color bar represents the percentage of RMS reduction.

**Table 3.** The RMS of residual time series before and after the PCA and ICA filtering.

| Direction | PCA | | | ICA | | |
|---|---|---|---|---|---|---|
| | Min (RMS) | Max (RMS) | Mean (RMS) | Min (RMS) | Max (RMS) | Mean (RMS) |
| E | 2.93% | 63.27% | 35.24% | 0.29% | 33.26% | 14.45% |
| N | 0.79% | 63.47% | 23.95% | 0.26% | 26.42% | 8.97% |
| U | 4.45% | 81.96% | 30.41% | 1.32% | 24.96% | 13.27% |

We calculate the velocities and uncertainties in the time series before and after PCA using the ICA filter (see Table S1). The results show that after applying the PCA filter, 79% (63 stations) of the horizontal velocities are within ±0.2 mm/year, 91% (72 stations) of the vertical velocities are within ±0.4 mm/year, and the associated speed uncertainties are reduced by 33.84%, 22.86%, and 26.59% in the E, N, and U components, respectively. In contrast, after applying the ICA filter, 98% (78 stations) of the horizontal velocities are within ±0.2 mm/year, 98% (78 stations) of the vertical velocities are within ±0.4 mm/year, and the associated speed uncertainties are reduced by 13.50%, 8.06%, and 11.82% in the E, N, and U components, respectively.

### 4.2. Noise Analysis After Applying ICA and PCA

Table 4 lists the mean noise results of 4 optimal models before and after filtering. Table 4 shows that the application of a regional filter can reduce the magnitudes of PN, FN, and GGM noise, especially in the U component, by approximately 20%. Upon further analysis, we find that the magnitude of WN is only 1–2 mm in all noise model combinations and can reach up to 3 mm in the U component at a few stations. The WN+FN+RW model is the optimal model only for stations MIN0 and PATN; at these stations, the magnitude of RW is approximately 10 mm in the E and N components and almost 0 in the U component. In contrast, the magnitude of RW at the stations with the WN+RW+GGM model as the optimal model are less than 1 mm in the E and U components and 2–3 mm in the N component.

The WN and RW magnitudes are basically consistent before and after PCA and ICA filtering, although the WN magnitude increases after PCA and ICA filtering at a few stations. We suppose that the balance between colored noise and WN is occasionally altered (some WN may have previously gone into biasing and thus increasing the amplitude of colored noise) and that WN is now more prominent and "increases". We can conclude that the magnitudes of WN in the residual time series are very small in Antarctica and that regional filters can reduce the magnitudes of PN, FN, and GGM but have little influence on those of WN and RW.

**Table 4.** The mean noise results of 4 optimal models before and after filter (mm).

| Noise | Direction | RAW | PCA | Reduce | RAW | ICA | Reduce |
|-------|-----------|------|-------|--------|-------|-------|--------|
|       | E | 4.83 | 2.62 | 44.16% | 4.90 | 3.82 | 21.69% |
| **1 \*** | N | 5.49 | 4.27 | 22.03% | 5.14 | 4.46 | 13.25% |
|       | U | 16.60 | 10.55 | 38.36% | 16.43 | 13.12 | 20.53% |
|       | E | 7.24 | 4.82 | 33.34% | 5.80 | 5.25 | 10.77% |
| **2** | N | 7.16 | 5.26 | 27.14% | 7.16 | 5.75 | 19.00% |
|       | U | 21.27 | 14.89 | 30.04% | 21.27 | 16.58 | 22.09% |
|       | E | 12.65 | 11.02 | 13.91% | 13.33 | 12.83 | 3.99% |
| **3** | N | 10.94 | 9.21 | 15.67% | 11.04 | 10.71 | 7.11% |
|       | U | 21.38 | 14.77 | 32.58% | 22.18 | 16.61 | 25.63% |
|       | E | 5.52 | 3.69 | 35.98% | 6.07 | 5.08 | 18.00% |
| **4** | N | 7.50 | 6.19 | 17.47% | 6.90 | 6.23 | 10.67% |
|       | U | 21.32 | 14.06 | 31.62% | 20.63 | 16.56 | 19.93% |

The raw and PCA/ICA are the mean magnitude of station time series before and after applying the PCA/ICA filter (the stations with consistent optimal noise models); the number of stations is different before and after applying the PCA and ICA filter, so there are two RAW. \* Noise combination 1 is the PN mean magnitude results of WN+PN, combination 2 is the FN mean magnitude result of WN+FN+RW, combination 3 is the GGM mean magnitude result of WN+RW+GGM, and combination 4 is the FN mean magnitude result of WN+FN. WN+RW appears only 1 time in the N component before and after applying PCA and 1 time in the U component before and after applying ICA; therefore, we have no statistics for the WN+RW model.

## 5. Conclusions

The CME and the optimal noise model are two of the most important factors affecting the accuracy of time series in regional GNSS networks. To obtain high-accuracy coordinate time series, we adopt factor analysis for the first time to explore the applicability of time series recorded by 79 GNSS stations in Antarctica from 2010 to 2018 and removed the CME of the residual time series by ICA filtering. The filtering results derived from PCA and ICA are compared and analyzed, after which the AIC is used to determine the optimal noise model before and after ICA/PCA filtering. The results show the following:

1.  After PCA filtering, the RMS values of the residual time series are reduced by 35.24%, 23.95% and 30.41% in the E, N, and U components, respectively, and the associated speed uncertainties are reduced by 33.84%, 22.86%, and 26.59%, respectively. Moreover, 79% of the horizontal velocities are within ±0.2 mm/year, and 91% of the vertical velocities are within ±0.4 mm/year. After ICA filtering, the RMS values of the residual time series are reduced by 14.45%, 8.97%, and 13.27% in the E, N, U components, respectively, and the associated speed uncertainties are reduced by 13.50%, 8.06% and 11.82%, respectively. Additionally, 98% (78 stations) of the horizontal velocities are within ±0.2 mm/year, and 98% (78 stations) of the vertical velocities are within ±0.4 mm/year. The PCA-extracted CME shows some variation over Antarctica, while the CME extracted using ICA has more obvious spatially uniform localized patterns, indicating that the CME derived from ICA performs better in Antarctica.

2.  Different GNSS time series in Antarctica have different optimal noise models with different noise characteristics in different components. The main noise models are the WN+FN and WN+PN models. Furthermore, the spectrum index of most PN is similar to that of FN. Regional filters

can reduce the magnitudes of PN, FN, and GGM but have little influence on those of WN and RW. Finally, there are more stations with consistent optimal noise models after ICA filtering than there are after PCA filtering.

**Supplementary Materials:** The following are available online at http://www.mdpi.com/2072-4292/11/4/386/s1.

**Author Contributions:** W.L. and S.Z. conceived and designed the experiments; W.L., J.L. and Q.Z. analyzed the data, F.L. gave the critical suggestions, W.L. wrote the main manuscript text, and the other authors helped with the writing of text.

**Funding:** This research was supported by the National Key Research and Development Program of China (2017YFA0603104), the State Key Program of National Natural Science Foundation of China under Grant (41531069), and Independent Scientific Research Program for Cross-disciplinary of Wuhan University (2042017kf0209).

**Acknowledgments:** The numerical calculations in this paper have been done on the supercomputing system in the Supercomputing Center of Wuhan University. We are thankful to Ming (Institute of Geospatial Information, Information Engineering University, China; National Key Laboratory of Geo-information Engineering, Xi'an Institute of Surveying and Mapping, China) for the intense discussion about data processing. The authors would also like to thank three anonymous reviewers and the editor Arlene. Cui, M. Sc. for their insightful comments and suggestions, which helped to improve the manuscript significantly.

**Conflicts of Interest:** The authors declare no conflict of interest.

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
