# Peer review of "Spatiotemporal Filtering and Noise Analysis for Regional GNSS Network in Antarctica Using Independent Component Analysis"

_remotesensing, doi:10.3390/rs11040386_

Round 1

Reviewer 1 Report

Dear Authors,

The research contributed to a better understanding how the CME factor affects the accuracy of time series in the Antarctic GPS networks. The topic presented is interesting and may revels improvement concerning the spatiotemporal filtering techniques of GPS time series.

To my opinion the paper is acceptable with Major Revision for publication in Remote Sensing Journal.

Remarks

- Line 18 - Why the observations are limited to 79 GPS stations?. There are many other publicly available geodetic observations for Antarctica (see Zanutta et al., 2018).

Zanutta, A.; Negusini, M.; Vittuari, L.; Martelli, L.; Cianfarra, P.; Salvini, F.; Mancini, F.; Sterzai, P.; Dubbini, M.; Capra, A. New Geodetic and Gravimetric Maps to Infer Geodynamics of Antarctica with Insights on Victoria Land. Remote Sens. 2018, 10(10), 1608. https://doi.org/10.3390/rs10101608.

- Line 35 - The current state of the research field should be reviewed, with more key publications cited. The Introduction has to be improved to be more comprehensible to scientists working outside the topic of the paper.

- Line 120 - In the 2.1 and 2.2 sections Equations and Formulas are mainly collected from Dong et al. 2006 [20] and from Liu et al. 2018 [23] but the descriptions are not clear as well as their significance and the contained parameters.

- Line 125 - The definition of covariance matrix is incomprehensible.

- Line 138 - What does "p" parameter means? What does " epsilon" parameter means?  

- Line 143 - The formula (7) is incomprehensible as regards the indices and not the definition of the contained symbols.

- Line 164 - What is "r"? "Indet(2p)" may be "ln(detC)".

- Line 176 - It would be good to insert the criterion first (equation 15) and then define how the likelihood L considered in it, is calculated.

- Line 186 - The symbols shown in Figure 1 are not clearly distinguished.

- Line 193 - Has the RegEM routine ever been adopted in similar contexts, to analyze GNSS series? Insert references and give explanation concerning this choice.

- Line 200 - Insert a legend more detailed in the caption of the Figure 2.

- Line 201 - Which software have you adopted to realize the PCA and ICA filtering?

- Line 214 - Figures 3 and 4 have to be commented in the manuscript furthermore the characters used are too small and cannot be read clearly.

- Line 231 - Figure 5 has to be commented in the manuscript furthermore the characters used are too small and cannot be read clearly.

- Line 233 - Section 3 is quite similar to Ming et al. 2017. [22] is incomplete in the References Section.

- Line 245 - Are Figure 6 to 11 all necessary ? These figures are too small, difficult to understand and poorly commented in the text.

- Line 268 - Local and regional geodynamic comments should also be done in a continental context, comparing the values obtained with those inferred from the bibliography.

- Line 309 - Based on which criteria the ICs are chosen? Why have you defined 77 ICs?

- Line 316 - The sentence is not verifiable because Figures are not readable.

- Line 334 - What is the real meaning of this figure? Enter a dedicated comment in the text.

- Line 336 - The caption of Table 1 need to be improved. Insert in the text dedicated comments concerning the procedure and the results.

- Line 357 - The sentence does not correspond to the caption of Figs. 13-15.

- Line 382 - Are the estimated velocities (Supplementary material, Table S1) obtained filtering CME significantly different from the raw ones? Can this analysis contribute to our better understanding of geodynamic trends in Antarctica? The authors do not comment on the broader importance of these findings.

- Line 395 - “before …applying regional filtering” is not showed. Clarify.

- Line 415 - Figure 17 has to be commented in the manuscript furthermore the characters used are too small and cannot be read clearly

- Lines 425-426 - It seems to me that ICA reduces uncertainty on rates less than PCA, this is in contrast to what is said on lines 22-23 of the abstract.

- Lines 428-430 - Where is shown what is said here? I do not think so in Table 4, perhaps in Fig. 13-15? So, why do not you talk about it when you show those figures? This sentence is also reported in the conclusions (lines 472-473) it would be nice to understand at which point of the work the results that prove this are shown.

- Line 447 - I don't understand what the RAW, PCA, RAW, ICA columns indicate. Why are there two RAW columns?

- Line 466 - The authors often use the term "spatially clustered pattern" as a synonym for "spatially uniform pattern" for CME, but "a cluster" suggests to me a non-uniform localized effect.

- Line 487 - All the References should be described following the Remote Sensing instructions (https://www.mdpi.com/journal/remotesensing/instructions)

Author Response

Dear reviewer,

We want to thank you for the extremely helpful comments provided for our paper. We have studied the valuable comments from you and tried our best to revise the manuscript. The point to point responds to the comments are listed as word file.

Reviewer 2 Report

This is a very interesting paper. Methods are adequately described and results quietly support the conclusions. Therefore, I recommend publication on Remote Sensing with minor revision. I have just a suggestion. I propose to improve the figures 3, 4 and 5, because they are not readable when the manuscript is printed. For detailed comments, please see attachment.

Author Response

Dear reviewer,

We want to thank you for the extremely helpful comments provided for our paper. We have studied the valuable comments from you and tried our best to revise the manuscript. The point to point responds to the comments are listed  the word file.

Reviewer 3 Report

The paper entitled “Spatiotemporal filtering and noise analysis for regional GNSS network in Antarctica using independent component analysis” presents a comparison between different models representing the noise in the GNSS stations. Particularly, authors compare five noise models: white noise (WN) + power law noise (PN), white noise (WN) + random walk noise (RW), white noise (WN) + flicker noise (FN), white noise (WN) + flicker noise (FN) + random walk noise (RW), and white noise (WN) + random walk noise (RW) plus generalized Gauss-Markov  (GGM). The topic presented in this paper is of interest for the remote sensing community, particularly, for that working on GNSS applications. In general, the paper is well written, and its structure helps the reader to understand the main contribution of this study. 

However, before acceptance to be published, authors need to clarify some statements mentioned in the manuscript. These statements are:

- For the non-uniform patterns (e.g. Figure 7), could the authors mention some the other factors that might be incorporated in future studies?

- Could the authors incorporate a frequency plot in Fig. 12 to help in the understanding of the spectral analysis?

- Could the authors include some comments about the robustness of the optimal noise model to be applied in other ecosystems?

Minor comments in the text are:

- The arrows are too small to be visualized in the figures 6-8

- The caption in figures needs to be at the same page as the figures.

- Is “Hector” an acronym of a system (see line 354, page 14)?

- In line 399, page 18, do authors refer to Table 3 instead of Table 2? Please, clarify.

Author Response

(The authors gave the same response as above.)

Round 2

Reviewer 1 Report

Dear Authors,

To my opinion the paper is now acceptable for publication in Remote Sensing Journal.